# Automated Classification of Online Sources for Infectious Disease Occurrences Using Machine-Learning-Based Natural Language Processing Approaches

**DOI:** 10.3390/ijerph17249467

**Published:** 2020-12-17

**Authors:** Mira Kim, Kyunghee Chae, Seungwoo Lee, Hong-Jun Jang, Sukil Kim

**Affiliations:** 1Department of Preventive Medicine, College of Medicine, The Catholic University of Korea, Seoul 06591, Korea; mkim7111@gmail.com (M.K.); Chae9999@gmail.com (K.C.); 2Department of Data and HPC Science, University of Science and Technology, Daejeon 34113, Korea; swlee@kisti.re.kr; 3Research Data Sharing Center, Korea Institute of Science and Technology Information, Daejeon 34141, Korea; hongjunjang@kisti.re.kr

**Keywords:** machine learning, infectious disease, public health surveillance, online document, classification

## Abstract

Collecting valid information from electronic sources to detect the potential outbreak of infectious disease is time-consuming and labor-intensive. The automated identification of relevant information using machine learning is necessary to respond to a potential disease outbreak. A total of 2864 documents were collected from various websites and subsequently manually categorized and labeled by two reviewers. Accurate labels for the training and test data were provided based on a reviewer consensus. Two machine learning algorithms—ConvNet and bidirectional long short-term memory (BiLSTM)—and two classification methods—DocClass and SenClass—were used for classifying the documents. The precision, recall, F1, accuracy, and area under the curve were measured to evaluate the performance of each model. ConvNet yielded higher average, min, and max accuracies (87.6%, 85.2%, and 91.1%, respectively) than BiLSTM with DocClass, while BiLSTM performed better than ConvNet with SenClass with average, min, and max accuracies of 92.8%, 92.6%, and 93.3%, respectively. The performance of BiLSTM with SenClass yielded an overall accuracy of 92.9% in classifying infectious disease occurrences. Machine learning had a compatible performance with a human expert given a particular text extraction system. This study suggests that analyzing information from the website using machine learning can achieve significant accuracies in the presence of abundant articles/documents.

## 1. Introduction

### 1.1. Infectious Diseases Public Health Threats

The outbreak of new infectious diseases, such as H7N9, H5N1, Zika, and Ebola, is a serious threat to human health and life. Some infectious diseases are associated with high mortality and fatality rates, and no treatment or vaccination is available for some of these diseases [1]. Therefore, disease monitoring is important to provide a prompt early warning of a potential disease outbreak.

The detection of abnormal disease distributions and the accurate assessment of the risk of disease outbreak is paramount [2]. Organizations at the national, state, and local levels have developed various disease surveillance systems to improve public health.

The prevention and management of infectious disease outbreaks is a growing concern due to the increased mobility of the population and frequent overseas travel. The Korea Disease Control and Prevention Agency (KDCA) is working on protecting the homeland from the transmission of infectious diseases from foreign countries.

One of the efforts that are being undertaken is to collect the outbreak information of the infectious diseases through credible international bodies and to abstract the core information required to detect potential disease occurrences. The daily reading or assimilation of a broad range and a large number of reports is labor-intensive, introduces variability between reviewers, and is hard to scale readily for large datasets. Thus, the automated identification of relevant information is the first step toward an effective response to a potential disease outbreak.

### 1.2. Online Sources of Infectious Disease Outbreaks

Information sources on the Internet, such as publicly available news media and social media, have been found to be informative for the early detection of emerging epidemics [3,4]. Over the years, several Internet-based biosurveillance projects have been launched to identify disease outbreak information from online articles.

For example, HealthMap was developed as a web-based surveillance tool for aggregating multiple online sources to monitor the outbreaks and display levels of disease risk on a map [3,5]. The Program for Monitoring Emerging Diseases (ProMED) is another one of the largest publicly available global disease reporting systems [6,7]. The Medical Information System (MedISys) monitors ProMED-Mail, web sites of national public health authorities, and news from around the world. It additionally monitors trends and determines alert levels for each disease and country by comparing the average numbers of recent news items to ensure a geographically balanced selection [8].

### 1.3. Machine Learning for Identifying the Outbreak of an Infectious Disease

Online news articles can provide timely information on disease outbreaks worldwide, though identifying relevant online articles is challenging due to the vast amount of ever-growing publications on the Web. Automated text classification using machine learning has been a promising approach to identifying online news articles relevant to disease outbreaks [9].

Machine learning classifiers, such as naïve Bayes, support vector machine (SVM), and bidirectional long short-term memory recurrent neural networks, have been applied to detecting online articles about infectious disease activities [9,10,11,12,13,14]. While several surveillance systems collect and disseminate surveillance information from online sources using text extraction, we specifically devised a system for classifying online documents using natural language processing and machine learning models.

The purpose of this study was to provide a quantitative evaluation of AI in the automated identification of information related to infectious disease occurrences using online sources.

## 2. Materials and Methods

### 2.1. Data Sources

This study was conducted by collecting data on the occurrence of infectious diseases from global websites, which were specified by the KDCA as important sources of public health information (Table 1).

We collected resources containing the names of 100 major infectious diseases. Five websites, including WHO-DON, WHO-IHR, WHO-AFRO, NCDC, and SAMOH, among the 10 sources yielded data that were mostly related to infectious disease occurrences, whereas ProMED and WHO-News contained several documents unrelated to disease occurrences. In order to select relevant documents, sources containing major infectious disease names were preferentially selected using the search function provided in the websites and the Google site designation search and then reviewed manually.

### 2.2. Data Preprocessing

A total of 2864 documents were collected using a web crawler. The documents were parsed into the title, body text, and metadata. The body text was then divided into sentences and normalized by unifying the publishing date format and deleting whitespace, newlines, comments, and superscripts.

Since we collected the information related to the current (at the time of publication) infectious disease occurrence, sources related to the past occurrence, general disease information, study results of infectious diseases, or the measures of treatments and prevention of infectious diseases were excluded.

The information about the occurrence of the infectious disease included the nature of the disease, the time of onset, the place of occurrence, and the person involved. Thus, the essential elements of a sentence for machine learning include the disease, person, event, place, and time. Disease refers to the name of the disease, diagnosis, or the pathogen causing the infectious disease, including bacteria, viruses, or fungi. Person refers to a subject affected by the disease. The event refers to the status of the person, such as suspected, probable, confirmed, infected, or dead. Place refers to the country, state or province, city, county, or location where the disease occurred. Time refers to the date and duration of the occurrence of the infectious disease.

### 2.3. Building the Gold Standard

Two research assistants read the title and the body text to identify the sources containing valid information regarding the occurrence of the target infectious diseases. Each reviewer independently labeled sources for inclusion if they carried information related to infectious disease occurrences, and excluded sources lacking such information. They were blinded to the label generated by the other reviewer. Conflicts were resolved by a medical doctor. Accurate labels for the training and test data were provided based on reviewer consensus.

### 2.4. Deep Learning Models

Determining the occurrence of infectious diseases in sources is a binary classification. We used two popular deep learning neural network models, namely, a character-level convolutional neural network (CNN) called ConvNet [15] and a word-level bidirectional long short-term memory (BiLSTM).

ConvNet treats each document as a sequence of characters, which is passed into six convolution and max-pooling layers and three fully connected layers to determine the probability that the document belongs to a positive class. This model learns rapidly and with reasonable performance compared to word-level models since it does not require a pre-trained embedded word.

Conversely, BiLSTM treats each document as a sequence of words, which is passed into two layers of a bidirectional LSTM, each of which is followed by dropout (0.5) and then the two fully connected layers to determine the probability that the report belongs to a positive class. To ensure reasonable performance, this model requires pre-trained word embedding vectors, which leads to slower learning compared to character-level models. We used 300D fasttext [16] for word embedding because it reduces the out-of-vocabulary (OOV) problem.

Using these two algorithms, we developed two classification methods, DocClass and SenClass, which represent document-level learning and sentence-level learning, respectively. DocClass classifies the document by using the entire document as input, and SenClass receives the document sentence as input, and classifies the sentence first, followed by the document. Figure 1 displays the basic framework of the model.

To train ConvNet and BiLSTM, the length of various documents must be fixed. Considering the constraint of hardware memory and the length distribution of the training data, the number of characters in the document was set to 10,000 in Conv-Net, and the number of words was set to 800 in BiLSTM for DocClass. In the case of SenClass, the number of characters in a sentence was set to 3200 in ConvNet, and the word limit was set to 128 in BiLSTM. Longer texts were truncated and shorter texts were padded. Furthermore, we set the learning rate to (0.001, 0.001) and batch size to (64, 128) for ConvNet and BiLSTM when we trained the two models. The structures of the two models are shown in Figure 2 and Figure 3, respectively.

### 2.5. Experimental Setup and Evaluation

The 2864 documents were randomly divided into 80% for training and 10% for validation. The remaining 10% was used for the model evaluation. The training data were used in model training, the validation data for performance evaluation during training, and the test data for the prediction and performance evaluation. The accuracy of the learning curve was investigated according to data size to determine the appropriateness of the training data size.

The statistical tests used to validate the performance of the proposed models were the area under the curve (AUC), accuracy, precision, recall, and the F1 score. AUC measures the effects of all possible classification thresholds and a numerical representation of the performance of the binary classifier, while the receiver operating characteristic (ROC) is the visual representation of the performance of the binary classifier.

Precision refers to the percentage of results that are relevant and recall refers to the percentage of total relevant results that are correctly classified by the algorithm. The F1 score represents the balance between the precision and recall scores. The F1 score is based on both the precision and recall of classification, and hence, is considered as a weighted average of the model precision and recall, with maximum and minimum values of 1 and 0, respectively.

## 3. Results

### 3.1. Manual Review of Documents

The number of documents collected for constructing a training dataset for each site is presented in Table 2. At the time of document collection, a total of 2864 documents retrieved from 10 websites were manually reviewed and classified into infectious disease occurrence and non-occurrence. Of the 2864 reports reviewed, 2083 (72.7%) were classified as relevant (disease occurrence) and 781 (27.3%) as irrelevant.

### 3.2. Performance Evaluation

The data originated from various sources and accordingly yielded different classification ratios, which were maintained when splitting the data into training, validation, and test categories, as shown in Table 3. The baseline accuracy of the test data was 73% when all the documents were classified as positive, i.e., they included an infectious disease occurrence. Regarding this experiment, we present the performance of two methods via document classification learning (DocClass) and sentence classification learning (SenClass). Accordingly, we trained two deep learning models (ConvNet and BiLSTM) with a mini-batch size = (32, 64, 128) and up to 100 epochs (using early stopping with patience = 40, 80) using training and validation data. We evaluated the models using test data.

The evaluation results are presented in Table 4. Due to the randomness of the deep learning models, we ran each experiment five times and obtained the average, standard deviation, maximum, and minimum accuracy values. In the case of DocClass being used on all documents, the ConvNet yielded a higher average, minimum, and maximum accuracies (88%, 85%, and 91%, respectively) than BiLSTM (84%, 83%, and 85%, respectively), while BiLSTM was a little more stable due to its smaller standard deviation.

The average accuracies of both models were significantly higher than the baseline accuracy of 73%. BiLSTM required additional training data compared with ConvNet to learn the sequence sufficiently. The used training data were not large enough, which might have decreased the accuracy of BiLSTM compared with ConvNet when using DocClass on all documents. However, in the case of using SenClass on all documents, BiLSTM yielded higher average, minimum, and maximum accuracies (93%, 93%, and 93%, respectively) than ConvNet (89%, 88%, and 91%, respectively), while BiLSTM was more stable since it has a smaller standard deviation. The performance of SenClass was better for both deep learning models than DocClass for all documents because the SenClass method contained more training datasets than the DocClass for adequate learning of the sequence.

The data contained very different ratios of inclusion by sources, and especially the data from ProMED and WHO-News were the hardest to classify. Thus, we additionally performed the same experiments with data derived from these two sources, and the results are presented in the right column of Table 4. The baseline accuracy of the data derived from ProMED and WHO-News was about 50%, whereas the deep learning models resulted in an accuracy of 71–72% (DocClass) and 83–88% (SenClass) on average, which was much higher than the baseline.

In the case of SenClass, when data were few and hard to classify, BiLSTM resulted in higher accuracy than ConvNet, while it still retained higher stability. These properties appeared to be due to the characteristics of the two models. ConvNet can use collocation within a short context but has difficulty in managing word sequences with long-term dependencies, while BiLSTM can detect long-term dependencies since it is a sequence-based model.

All the documents from the hardest sources, i.e., ProMED and WHO-News, carried some of the essential elements, i.e., at least infectious disease names, indicating disease occurrence. However, only about half of the documents contained appropriate sequences of words corresponding to the essential elements and were classified as an infectious disease occurrence.

In many of the sequences of words, the occurrence of essential elements was not limited to a short context, which might increase the failure of ConvNet to correctly classify documents from the hardest sources. Conversely, BiLSTM facilitated the detection of long-term dependencies between the essential elements since it deals with the whole sequence of words in documents.

All performance results for the deep learning models are listed in Table 5. The performance of BiLSTM with SenClass yielded an overall accuracy of 92.9% when classifying infectious disease occurrences. The precision, recall, and F1 measure were also strongly similar to accuracy because the task involved binary classification and we used the learning policy called early stopping in which the model stopped learning when it achieved the highest F1; the highest precision and recall usually occur at the same point.

Figure 4 shows the corresponding ROC curves when using DocClass and SenClass with the two models with all test documents (ConvNet-All and BiLSTM-All) and the hard test documents from hard sources (ConvNet-Hard and BiLSTM-Hard). In the case of using DocClass on all test and hard test documents, ConvNet was substantially better than BiLSTM in the left-upper corner. In the case of using SenClass on all test documents, BiLSTM was better than ConvNet in the left-upper corner. However, BiLSTM was similar to ConvNet in the case of using SenClass on hard test documents.

We compared the BiLSTM-based SenClass with three baseline machine learning classifiers, namely, Gaussian naïve Bayes, linear SVM, and random forest. Similar to SenClass, these three baseline methods also classify documents by classifying sentences. Table 6 shows that the BiLSTM-based SenClass outperformed the three baseline methods in core measures (accuracy, F1, and AUC). The three baseline methods extract feature vectors based on TF-IDF (Term Frequency-Inverse Document Frequency) through pre-processing, such as tokenization, stopword removal, and stemming for each sentence. We used cross-validation and a grid search to tune the hyperparameters of the three baseline methods and train the models. Therefore, the three baseline methods merged the training and validation data and used the merged data for training. In particular, due to the high computational cost of the linear SVM according to the number of sentence samples compared to other models, only 10% of merged data were used for training.

### 3.3. Learning Curve Analysis

To determine whether the size of the training dataset was adequate, we drew a learning curve for the accuracy according to the increase in the size of the training data, as shown in Figure 5. We trained ConvNet using 20%, 40%, 60%, 80%, and 100% of the training data from all sources and calculated the accuracy of the two methods using the test data. We maintained the document ratio from each source in all subsets of the training data.

The accuracy of DocClass still increased even though the increased ratio declined after 40% of the training data was used. The accuracy of SenClass, however, was not significantly affected by the size of the training dataset. However, the lowest-performing accuracy of SenClass was similar to the highest degree of accuracy when using DocClass. In addition, the accuracy of SenClass increased slightly.

We found that the character-based ConvNet classifier of DocClass performed reasonably well, with an 87.6% accuracy and a 95.1% AUC, but often failed to correctly classify documents from the hardest sources. The ConvNet classifier using SenClass performed well, with an 89.4% accuracy and a 94.9% AUC, and showed higher accuracy and AUC than DocClass, even from the hardest sources.

BiLSTM using SenClass showed potential for document classification from the hardest sources. We also observed very low standard deviations (SDs) for the performance measures in the BiLSTM using both DocClass and SenClass, which suggests that BiLSTM was more stable and less sensitive to the initial parameter setting when training using the classification of documents containing references regarding infectious disease occurrences.

## 4. Discussion

In this pilot study, the feasibility and accuracy of machine learning techniques were determined using automated natural language processing to classify infectious disease occurrence information online. It is labor-intensive and time-consuming to collect information regarding infectious disease outbreaks and to abstract the core information needed to determine the potential disease occurrence. Thus, an automated classification approach using machine learning to diminish the manual workload was explored in this study.

### 4.1. Principal Findings

Our natural language processing (NLP)-based machine learning technique facilitated the classification of information about infectious disease occurrences from a large volume of online sources. The healthcare field is an area of application for machine learning since it contains vast data resources that are difficult to handle manually. The machine learning classifier in this study was not entirely accurate; however, the accuracy was comparable or superior to other studies in the healthcare field, which is mainly focused on disease diagnosis [17,18], the prediction of disease risk [19,20,21], and the classification of disease [22,23].

The actual accuracy of our classifier is expected to be higher than the study results when applied in the real world because the news related to infectious disease occurrence is posted on many relevant websites with similar content. Therefore, even if one document is incorrectly classified on one site, the possibility that the same document is correctly classified on other sites is high.

Deep learning models are hindered by the need for a large training dataset, and the accuracy of the algorithm largely depends on the size of the training set based on experience in other domains [24]. The performance of recent studies (i.e., SST-2 [25] and Yelp-2 [26]) in binary text classification [27] was better than our study. However, SST-2 and Yelp-2 carried a significantly larger number of training data than our study. Nevertheless, in the case of a task involving infectious disease occurrence classifications, the results showed that deep learning models can perform well, even with a training dataset comprising only 2864 labeled documents. Thus, we believe that the performance of the classifier will improve further with larger datasets since the learning curve of accuracy grew as the size of the training dataset increased, as shown in Figure 5.

Furthermore, the task was harder than other binary text classifications, such as sentiment analysis of a review text. Our task data was quite long with each text exceeding 800 words on average and a 9000-word maximum. Important clue words (essential elements) indicating the occurrence of the infectious disease usually appeared within one to two sentences, suggesting the need for a different approach to classifying each sentence first, and then merging the results of each sentence classification to classify the document, instead of classifying documents directly. Therefore, we developed SenClass to classify documents by learning in sentence units, where SenClass showed better performance than DocClass, a document-unit learning classifier.

### 4.2. Limitations and Future Directions

Some limitations at the current stage need to be addressed in future work. We extracted text only from the websites, which provided documents in an HTML format, and extracted text through a pdf reader if documents were provided in a pdf format. When extracting text from a pdf document, structural information was removed and the strings were extracted in the order they appeared on the screen. As a result, the original order of words was distorted and the sentence boundary was unclear. There is a risk of generating a relatively large number of errors in constructing a training dataset.

We only focused on English document sources in this study. However, due to the globalization and mobility of infectious diseases, some important and timely news, especially related to disease outbreaks, may be reported in other languages. Future studies should incorporate a multilingual processing component to deal with important news sources in other languages.

In the current deep learning experiments, we did not use additional features (entities), such as disease, person, place, and time, which are essential elements to determine whether a report mentions an infectious disease outbreak. In a future study, we will plan to first identify such entities from reports and elucidate the role of the additional features in deep learning models and their performance.

## 5. Conclusions

We proposed a general framework for building an automated, infectious disease-specific online document collection and classification system, which is becoming increasingly critical for protecting public health. Such a framework can automatically gather specific infectious-disease-related documents from the Web and provide accurate and timely information to health providers.

This study demonstrated the feasibility of using machine learning techniques to identify infectious disease outbreaks. The artificial intelligence is compatible with the performance of a human expert given a specific text extraction system. This study suggests that collecting and classifying online documents using machine learning is substantially accurate in the presence of a plethora of resources on the Internet.

## Figures and Tables

**Figure 1 ijerph-17-09467-f001:**
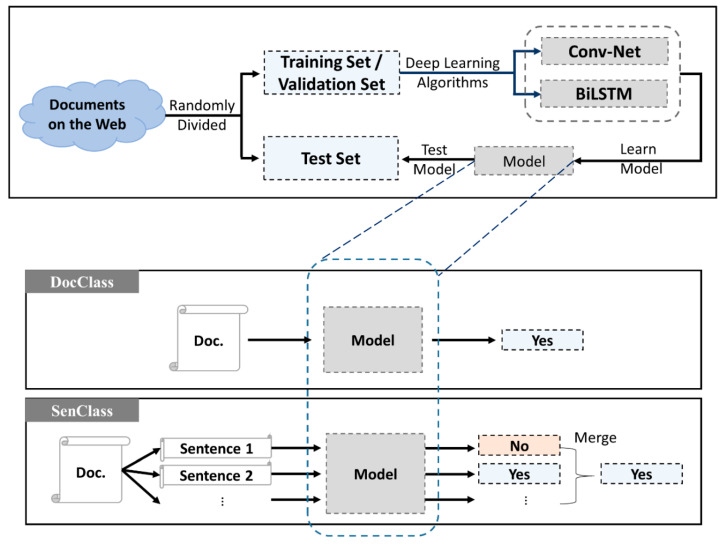
The two machine learning algorithms and classification methods used in the infectious disease occurrence detection experiments. BiLSTM: bidirectional long short-term memory.

**Figure 2 ijerph-17-09467-f002:**
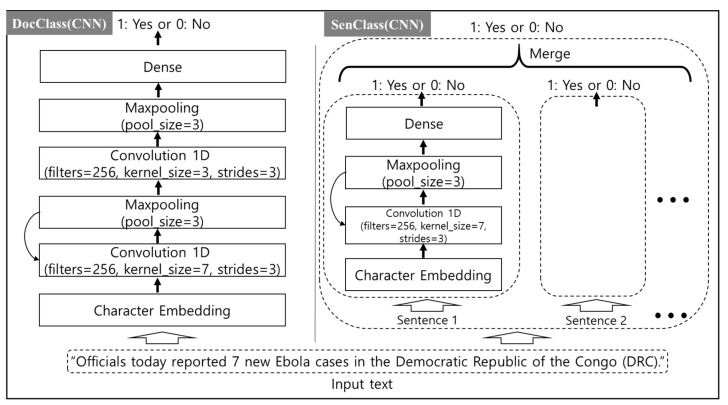
The ConvNet model structure for two classification methods, DocClass and SenClass. CNN: convolutional neural network.

**Figure 3 ijerph-17-09467-f003:**
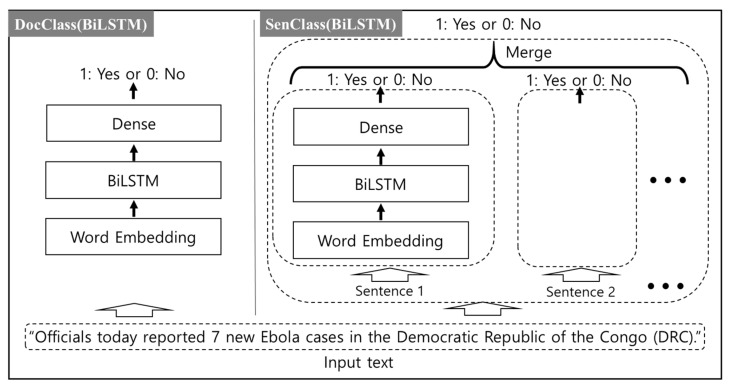
The BiLSTM model structure for two classification methods, DocClass and SenClass.

**Figure 4 ijerph-17-09467-f004:**
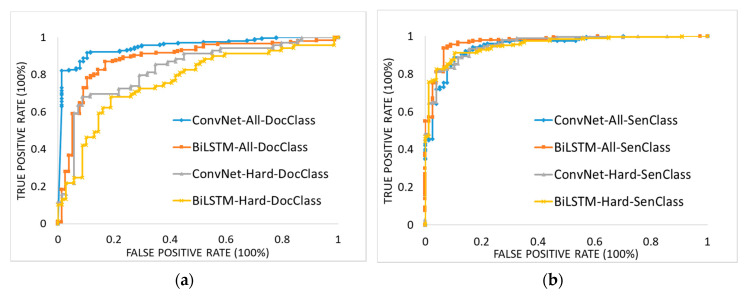
Receiver operating characteristic (ROC) curves of models classifying infectious disease occurrences: (**a**) ROC curves using DocClass for ConvNet and BiLSTM models and (**b**) ROC curves using SenClass for ConvNet and BiLSTM models.

**Figure 5 ijerph-17-09467-f005:**
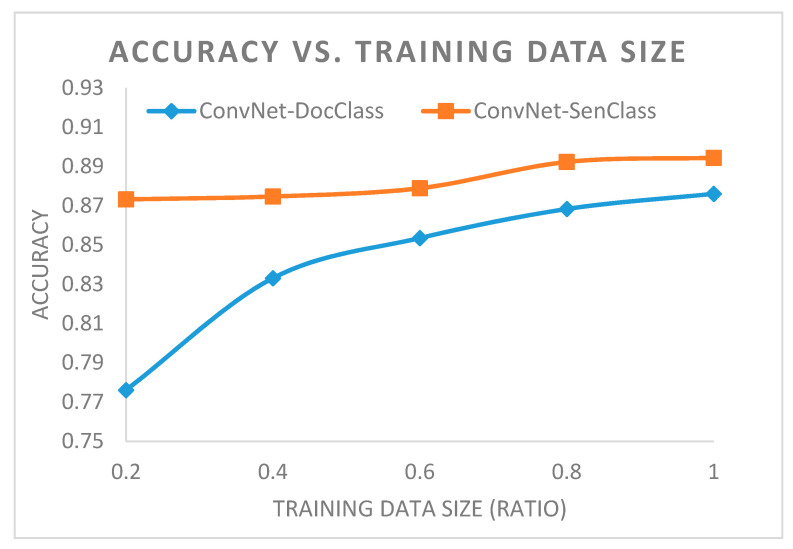
Learning curve.

**Table 1 ijerph-17-09467-t001:** List of websites for data collection.

Website	URL
CIDRAP	http://www.cidrap.umn.edu/
HealthMap	https://www.healthmap.org/
Medisys	https://medisys.newsbrief.eu/
NCDC	https://ncdc.gov.ng
ProMED	http://www.promedmail.org/
SAMOH	https://www.moh.gov.sa/en/CCC/
WHO-AFRO	http://www.afro.who.int/health-topics/disease-outbreaks/outbreaks-and-other-emergencies-updates
WHO-DON	https://www.who.int/csr/don/en
WHO-IHR	https://apps.who.int/ihr/eventinformation
WHO-News	https://www.who.int/news-room

**Table 2 ijerph-17-09467-t002:** Gold standard for identifying infectious disease occurrences.

Sources	No. of Documents	Gold Standard		
		Inclusion	Exclusion	Ratio
CIDRAP	51	50	1	98%
HealthMap	4	3	1	75%
MedISys	22	20	2	91%
NCDC	30	13	17	43%
ProMED	159	159	0	100%
SAMOH	97	97	0	100%
WHO-AFRO	1000	939	61	94%
WHO-DON	116	106	10	91%
WHO-IHR	385	152	233	39%
WHO-News	1000	544	456	54%
Total	2864	2083	781	73%

**Table 3 ijerph-17-09467-t003:** Training, validation, and test data.

Source	Total	Training				Validation				Test			
		Count	Inclusion	Exclusion	Ratio	Count	Inclusion	Exclusion	Ratio	Count	Inclusion	Exclusion	Ratio
CIDRAP	51	41	40	1	98%	5	5	0	100%	5	5	0	100%
HealthMap	4	4	3	1	75%	0	0	0	0%	0	0	0	0%
Medisys	22	18	16	2	89%	2	2	0	100%	2	2	0	100%
SAMOH	30	24	12	12	43%	3	0	3	0%	3	1	2	33%
NCDC	159	127	127	0	100%	16	16	0	100%	16	16	0	100%
WHO-AFRO	97	79	79	0	100%	9	9	0	100%	9	9	0	100%
WHO-DON	1000	800	752	48	94%	100	92	8	92%	100	95	5	95%
WHO-IHR	116	94	86	8	91%	11	10	1	91%	11	10	1	91%
WHO-News	385	309	124	185	40%	38	14	24	37%	38	14	24	37%
ProMED	1000	800	435	365	54%	100	54	46	54%	100	55	45	55%
Total	2864	2296	1674	622	73%	284	202	82	71%	284	207	77	73%

**Table 4 ijerph-17-09467-t004:** The accuracy of the deep learning models for detecting infectious disease occurrences.

Models	Measures	All Documents		ProMED and WHO-News Documents Only (Hard)	
		BiLSTM	ConvNet	BiLSTM	ConvNet
DocClass	Run#1	0.827465	0.866197	0.717391	0.673913
	Run#2	0.838028	0.852113	0.702899	0.681159
	Run#3	0.852113	0.873239	0.717391	0.760870
	Run#4	0.834507	0.911972	0.702899	0.782609
	Run#5	0.845070	0.876761	0.717391	0.695652
	Average	0.839437	0.876056	0.711594	0.718841
	Std Dev	0.009514	0.022186	0.007938	0.049520
	Min	0.827465	0.852113	0.702899	0.673913
	Max	0.852113	0.911972	0.717391	0.782609
SenClass	Run#1	0.926056	0.894366	0.869718	0.830986
	Run#2	0.933099	0.901409	0.862676	0.788732
	Run#3	0.926056	0.880282	0.901409	0.806338
	Run#4	0.933099	0.883803	0.894366	0.852113
	Run#5	0.926056	0.911972	0.880282	0.855634
	Average	0.928873	0.894366	0.881690	0.826761
	Std Dev	0.003857	0.012937	0.016251	0.028972
	Min	0.926056	0.880282	0.862676	0.788732
	Max	0.933099	0.911972	0.901409	0.855634

**Table 5 ijerph-17-09467-t005:** Comparison of ConvNet and BiLSTM based on the precision, recall, F1, accuracy, and area under the curve (AUC) measurements.

Models	Sources	Machine Learning	Precision	Recall	F1	Accuracy	AUC
DocClass	All documents	ConvNet	0.885276	0.876056	0.878662	0.876056	0.9506
		BiLSTM	0.836505	0.839437	0.836018	0.839437	0.8829
	ProMED and WHO-News only (hard)	ConvNet	0.731487	0.718841	0.715217	0.718841	0.8318
		BiLSTM	0.714365	0.711594	0.710630	0.711594	0.7652
SenClass	All documents	ConvNet	0.898752	0.894366	0.895760	0.894366	0.9491
		BiLSTM	0.929231	0.928873	0.928864	0.928873	0.9706
	ProMED and WHO-News only (hard)	ConvNet	0.864100	0.826761	0.834486	0.826761	0.9568
		BiLSTM	0.887245	0.881690	0.883484	0.881690	0.9547

**Table 6 ijerph-17-09467-t006:** Comparison of SenClass and the three baseline methods based on the precision, recall, F1, accuracy, and AUC measurements.

Machine Learning	Sources	Precision	Recall	F1	Accuracy	AUC
Gaussian Naïve Bayes	All documents	0.884259	0.922705	0.903073	0.855634	0.8476
Linear SVM	All documents	0.862661	0.971014	0.913636	0.866197	0.9391
Random Forest	All documents	0.987805	0.782609	0.873315	0.834507	0.9605
SenClass_BiLSTM	All documents	0.929231	0.928873	0.928864	0.928873	0.9706

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
