# Peer review of "Automated Classification of Online Sources for Infectious Disease Occurrences Using Machine-Learning-Based Natural Language Processing Approaches"

_ijerph, 2020, doi:10.3390/ijerph17249467_

Round 1
Reviewer 1 Report
The present study provides an Effect Automated online sources classification for infectious disease occurrence using machine learning-based natural language processing approaches.
There are some typing and grammar issues In the manuscript. I suggest making it review again, with a native English speaker.
I have two general comments:
- The introduction needs more substantial support for the hypothesis. I could see there is a lack of references between epidemiology and temporal resolutions on the studies presented.
- I could not find the justifications of the sources used in the study. And this is important to be established.
I have some specific comments:
Lines 146 - 148 The description of the Precision and Recall becomes repetitive.
Line 161 - Repetitive using "Different" the use of synonyms is suggested.
Line 162 - Inappropriate use of punctuation in "... validation, and ..."
Table 3 - Check caption, reference the source or author. - Inappropriate table format in "Includ-e" and "Exclud-e"
Page 1-12. Using "93%, 93%, 93%", improve wording 13 to 20 - Improve writing, repetitive.
Reviewer 2 Report
This paper propose to use deep learning for building an automated, infectious disease-specific online document classification system. As an interesting topic, the paper is generally well written and easy to follow. There are some comments that could potentially improve the manuscript further:
1, Relevant literature on the use of machine learning with application to the outbreaks of infectious disease is completely missing.
2, Section 2.4 should be more specific on the two deep neural networks such as the filters, stride size, learning rate and any other detail that could potentially affect the performance. Consider drawing two structures while adding those details.
3, In addition to the baseline set manually, the paper should try other baseline methods commonly used in natural language processing such as the Naive Bayes, with the aim to get a comprehensive picture of its performance.
4, it is not clear of the specific gold standard used to manually categorise the collected documents. Obviously, such manual labelling that is highly labour-intensive does not make the proposed framework fully automated, as it claims 'We propose a general framework for building an automated, infectious disease-specific online document collection and classification system'
